# Comparative Study of Geological Hazard Evaluation Systems Using Grid Units and Slope Units under Different Rainfall Conditions

**Shuai Liu, Jieyong Zhu \*, Dehu Yang and Bo Ma**

Faculty of Land and Resources Engineering, Kunming University of Science and Technology,
Kunming 650032, China
\* Correspondence: zhujieyong@kust.edu.cn

**Abstract:** The selection of evaluation units in geological hazard evaluation systems is crucial for the evaluation results. In an evaluation system, relevant geological evaluation factors are selected and the study area is divided into multiple regular or irregular independent units, such as grids, slopes, and basins. Each evaluation unit, which includes evaluation factor attributes and hazard point distribution data, is placed as an independent individual in a corresponding evaluation model for use in a calculation, and finally a risk index for the entire study area is obtained. In order to compare the influence of the selection of grid units or slope units—two units frequently used in geological hazard evaluation studies—on the accuracy of evaluation results, this paper takes Yuanyang County, Yunnan Province, China, as a case study area. The area was divided into 7851 slope units by the catchment basin method and 12,985,257 grid units by means of an optimal grid unit algorithm. Nine evaluation factors for geological hazards were selected, including elevation, slope, aspect, curvature, land-use type, distance from a fault, distance from a river, engineering geological rock group, and landform type. In order to ensure the objective comparison of evaluation results for geological hazard susceptibility with respect to grid units and slope units, the weighted information model combining the subjective weighting AHP (analytic hierarchy process) and the objective statistical ICM (information content model) were used to evaluate susceptibility with both units. Geological risk evaluation results for collapses and landslides under heavy rain (25–50 mm), rainstorm (50–100 mm), heavy rainstorm (150–250 mm), and extraordinary rainstorm (>250 mm) conditions were obtained. The results showed that the zoning results produced under the slope unit system were better than those produced under the grid unit system in terms of the distribution relationship between hazard points and hazard levels. In addition, ROC (receiver operating characteristic) curves were used to test the results of susceptibility and risk assessments. The AUC (area under the curve) values of the slope unit system were higher than those of the grid unit system. Finally, the evaluation results obtained with slope units were more reasonable and accurate. Compared with the results from an actual geological hazard susceptibility and risk survey, the evaluation results for collapse and landslide geological hazards under the slope unit system were highly consistent with the actual survey results.

**Keywords:** susceptibility evaluation; risk evaluation; grid unit; slope unit; weighted information content method; rainfall

## 1. Introduction

Geological hazards refer to geological activities or geological phenomena that occur under the action of natural or human factors and which cause loss of human life and property and damage to the environment [1]. The selection of evaluation units is the core part of the process of geological hazard evaluation, and different evaluation units will influence the scientificity, accuracy, and feasibility of evaluation results. Commonly used evaluation unit types are grid units, sub-basin units, slope units, etc. [2]. Sub-basin units are mainly used to evaluate debris flow hazard susceptibility zoning [3]. As the

two most frequently used evaluation units in geological hazard evaluation systems, grid units and slope units each have their advantages and disadvantages. The slope unit is the basic unit used in assessments of the development of geological hazards, such as landslides and collapses. Terrain factors, such as slope, aspect, and height differences, have obvious controlling effects on the formation of slopes. Slope structure basically refers to the hydrogeological conditions, and the evaluation factors reflect the basic characteristics of slopes. As the basic unit of landslide development, the slope unit takes topography into account, but the differences between geological environments in different regions are not considered in a single unit, and the calculation speed in such analyses is slow [4,5]. The grid unit is not practical for areas with large variations in terrain and complex geological structures and it cannot fully reflect the terrain environment; however, the calculations involved are simple and convenient, and the system still has certain advantages for areas with large numbers of units [4–7]. At present, the most outstanding scholars use grid units as the evaluation units. Kunlong Yin et al., Shujun Tian et al., and Lei Wang et al. used grid units to evaluate susceptibilities to geological hazards [4–6], while Chenglong Yu, Haomeng Zhu et al., and Xiaoyan Zhao et al. used slope units to evaluate the same [7–9]. How to select the evaluation unit is a key problem in geological hazard evaluation.

In order to fully reflect the characteristics of grid units and slope units and ensure the objectivity and comparability of geological disaster evaluation results under the two evaluation systems, the selection of an evaluation model is particularly important. Now, due to the original data processing methods being different, weight determination methods for different evaluation factors can be divided into two categories: subjective and objective weighting methods. The supervisor weighting method mainly relies on personal experience to assign weights and is characterized by a high degree of subjective arbitrariness, while the objective weighting method, though based on mathematical theory, relies too much on objective data and ignores the determination of the weight of each evaluation factor. Common weight determination methods include the information quantity method, the analytic hierarchy process, the entropy weight method, the random forest method, logistic regression, and other machine-learning models [10–21]. Given this range of methods, outstanding scholars choose combinations of subjective and objective and qualitative and quantitative methods to give weights to each evaluation factor. Qi Qi et al., Zhongyuan Zhang et al., and Jing Yao et al. adopted the information quantity method combined with hierarchical analysis to carry out evaluations of geological hazard susceptibilities [22–25], and Libing Gao et al. adopted the method of entropy weighting and hierarchical analysis to carry out an evaluation of geological hazard susceptibility [26]. A. Małk established a landslide susceptibility evaluation model for Gdynia City in Poland by means of a landslide index, weight of evidence, and a logistic regression method and analyzed the significance of the evaluation factors with respect to landslide risk [27]. B. Zeng et al. established a prediction model of the spatial distribution of landslides using artificial neural networks by selecting an evaluation index system representing the factors affecting the stability of the Silurian slope in Enshi, Hubei Province, and finally verified the results using remote sensing data and field investigations [28].

In order to ensure that geological hazard risk evaluation results under the two evaluation unit systems involving grid units and slope units are reasonable, accurate, objective, and comparative, the geological hazard risk evaluation method is also important. Rainfall, as an important geological hazard inducing factor, plays an important role in risk evaluations of geological hazards. However, regarding susceptibility to geological hazards, the division of risk is relatively vague. Most existing studies take rainfall as an inducing factor for use in geological hazard risk evaluation, which leads to the generation of irrelevant risk evaluation results. The use of this indicator cannot better guide the accurate implementation of geological hazard prevention and mitigation. With respect to this issue, Yongtao Ji et al. took annual maximum daily rainfall at different frequencies as a hazard trigger factor and carried out a geological hazard risk evaluation based on the assumption of extreme rainfall [26].

In summary, taking Yuanyang County, Yunnan Province, China, as an example, this paper carries out a comparative evaluation of susceptibilities to geological hazards of collapses and landslides using two evaluation units: grid units and slope units, based on the principle of fully presenting the characteristics of grid units and slope units, with control variables as a basis for comparison, using the weighted information method combining the information method and the hierarchical analysis method. At the same time, on this basis, the disadvantage of rainfall as a single inducing factor in the evaluation system of geological hazard risk is improved. Maximum annual average daily rainfall was taken as the inducing factor, and grid units and slope units were used as the evaluation units to carry out risk evaluations of the geological hazards of collapses and landslides under four different rainfall conditions: heavy rain, rainstorm, heavy rainstorm, and extraordinary rainstorm conditions. Through data comparison and model accuracy comparison, it was found that the risk assessment results for the four rainfall conditions under the slope unit system were more reasonable and accurate and that the evaluation results were highly consistent with actual survey results. Therefore, with the aim of providing a scientific basis for geological hazard prevention and control, the susceptibility and risk zoning results obtained using the two evaluation units were compared and evaluated to solve the problem of selecting evaluation units for large area evaluations.

## 2. Overview of the Research Area

Yuanyang County is located in the southern section of the Ailao Mountain range in the south of Yunnan Province, China, on the south bank of the Honghe River in the southwest of Honghe Prefecture. It is located between $102°27'–103°13'$ E and $22°49'–23°19'$ N. It is bordered by Jinping County in the east, Luchun County in the south, Honghe County in the west, and Jianshui County, Gejiu City, and Mengzi County across the Red River in the north, spanning 74 km from east to west and 55 km from north to south, with a land area of 2189.88 square kilometers [21]. It is located in the southern section of the Ailao mountain range, with high mountains and deep valleys and vertical and horizontal gullies. It belongs to the type of deep cut middle mountain landforms. The county is north of the Red River and south of the deep Tengtiao River system; in a "V"-shaped development, the center of the formation is high and the sides are low, and the terrain from northwest to southeast is sloping. The highest altitude is 2939.6 m above sea level in Baiyanzi Mountain, and the lowest altitude is 144 m at the mouth of the small river estuary on the border with Jinping, with a relative height difference of 2795.6 m. The foundation of the regional paleogeomorphology was laid during the Indosinian Movement, and the prototype of the modern landform was formed at the end of the Yanshan Movement. Since the Holocene, river formation and valley cutting have given rise to a steep valley with a broken shape; the formations of the north and south parts have different modern topographies, showing terrain slope, slope shape, and slope variability. After the Hualixi Movement and the Himalayan orogen, four basic geomorphic units, erosion/denudation low mountain geomorphology, karst medium mountain geomorphology, accumulation valley geomorphology, and tectonic erosion medium mountain geomorphology, were formed in the region. The geological structure in the area is complex, and the neotectonic movement is strong. The geological structural features are mainly faults, followed by folds. The controlling structures are the Ailaoshan superlithic fault and the Red River crust fault in the NW-SE direction, with a series of folds. The Proterozoic Ailao Mountain Group Ptaa, Ptab, Ptf and Ptw are the most widely distributed strata in Yuanyang County. The weathering resistance of rock mass is weak, the rock strata are broken, and cracks have developed. Sloping land reclamation, infrastructure construction, and mining of mineral resources are the most important human engineering activities at present. From the analysis of the geological environmental conditions of the whole county, the adverse geological effects in Yuanyang County include karst, river valley erosion, goaf, soil erosion, etc., resulting in geological hazards, including collapse, landslide, debris flow, etc. Geological hazards are strongly developed and multi-point, wide-ranging and intensive, occurring in

groups and causing short-term disasters and serious loss. According to the field survey, Yuanyang County has a total of 228 collapse and landslide geological hazards. Among them, 201 were landslides, 21 were unstable slopes, and 6 were collapses (Figure 1). In general, the landslide hazard is more prominent and serious. The dense zone of geological hazards is consistent with the distribution area of the regional stress field, which is mostly developed in the zone between the two active faults of Honghe and Ailaoshan. Geological hazards mostly develop in slope areas with altitudes of 1000–2000 m and terrain slopes of 15°–40°. From the distribution of hazard points in each township (town), Xinjie Town has the most with a total of 50, accounting for 15.02% of the total number of hazards. According to the scale of geological hazards, 228 are mainly small geological hazards, followed by medium-sized hazards, and large-scale geological hazards are relatively few.

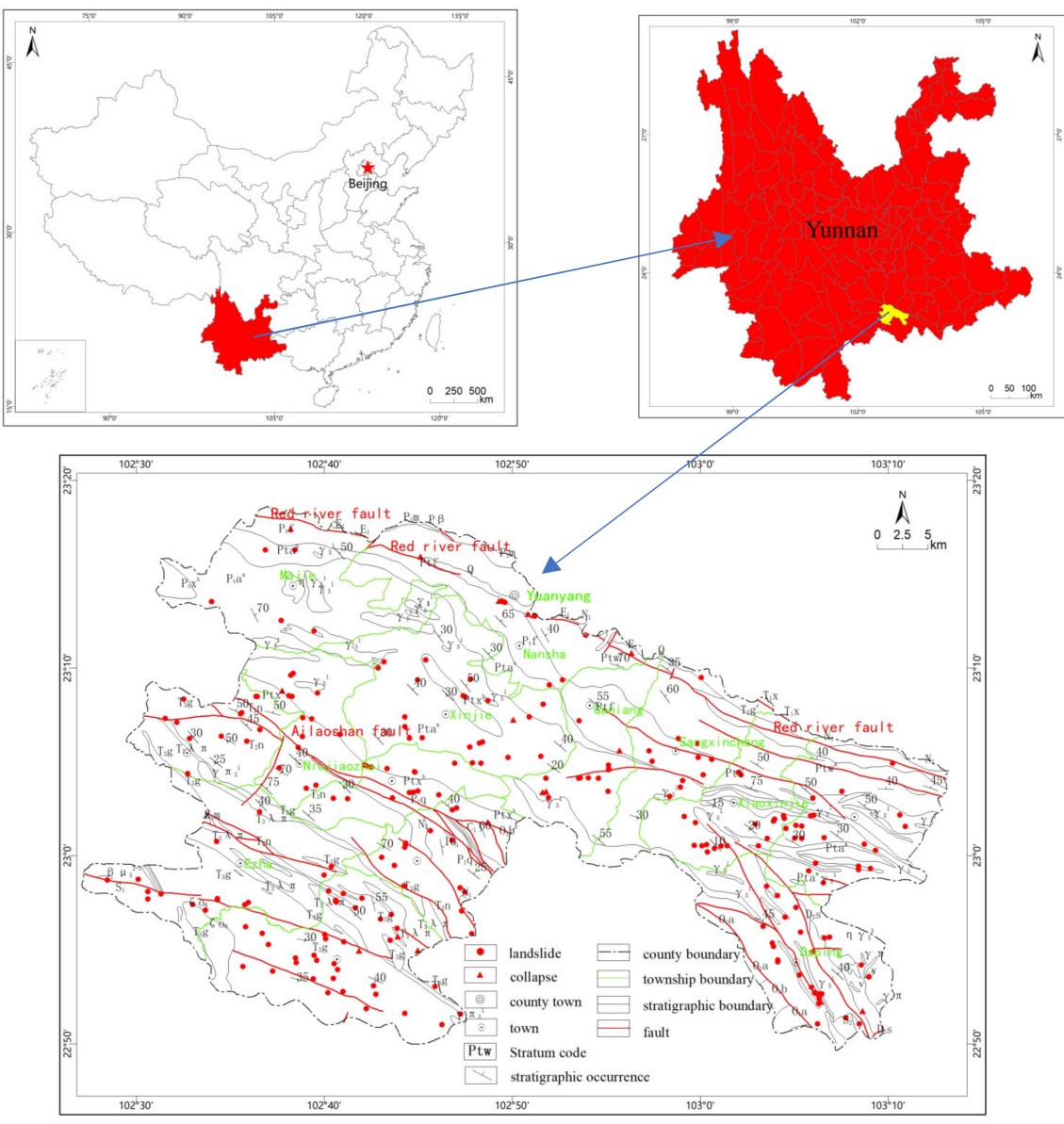

**Figure 1.** Location of the study area and distribution of hazard points.

## 3. Materials and Methods of Evaluation

Based on the GIS (Geographical Information System) platform, this paper establishes a geological hazard evaluation system for the purpose of objectively and accurately comparing geological hazard evaluation results under the grid unit and slope unit systems (Figure 2).

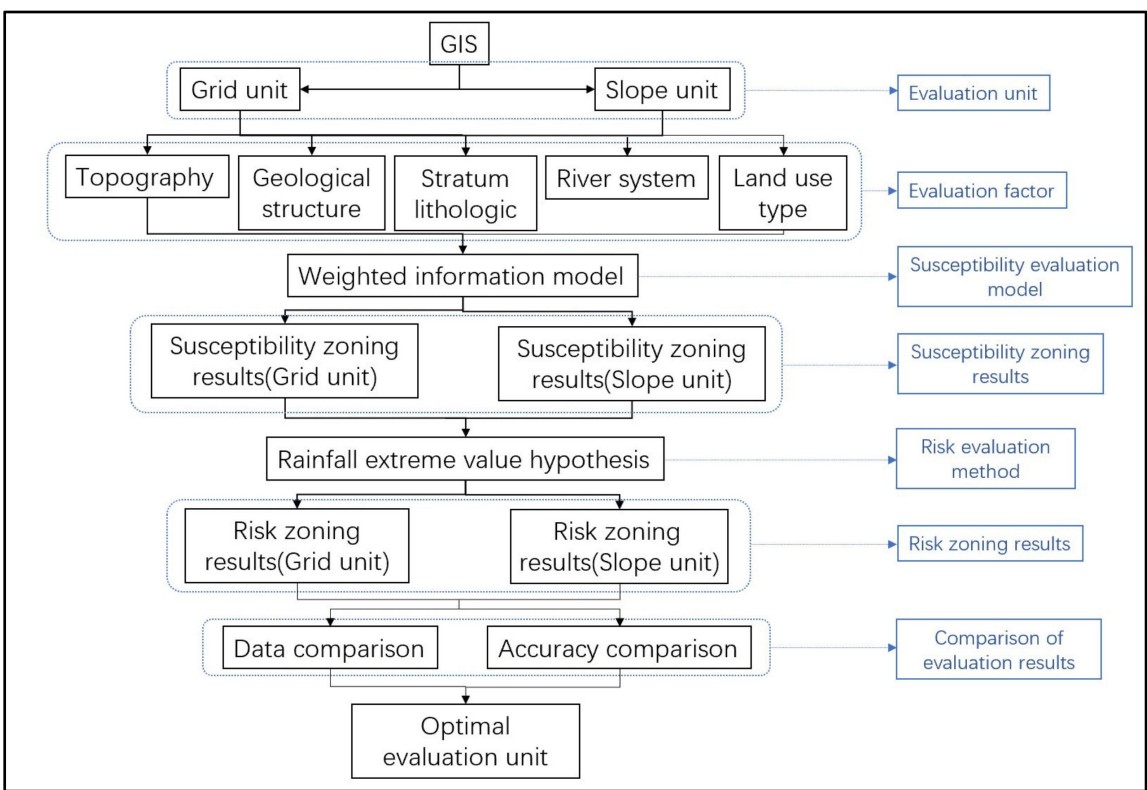

**Figure 2.** Evaluation system establishment process.

### 3.1. Establishment of Evaluation Method

In this paper, the susceptibility and risk index for each evaluation unit in the study area was calculated as a measure of the degree of geological hazards. The specific calculation formula is as follows.

#### 3.1.1. Establishment of Susceptibility Evaluation Method

(1)  ICM (Information Content Method)

The information content method is proposed on the basis of information theory and the engineering geology analogy method [6]. This method calculates the information value for each evaluation factor on the basis of the known information regarding collapses and landslides and the evaluation factor. The greater the value, the stronger the correlation between the evaluation factor and geological hazard. The amount of information (*I*) contributed by the secondary factor ($x_i$) of each evaluation index to the collapse and landslide hazard (*H*) can be calculated using the following formula [23]:

$$I\left(x_{i,}\ H\right) = ln\ \frac{N_i/N}{S_i/S} \tag{1}$$

In the formula, *N* is the total number of hazard points; *S* is the total number of evaluation units; $N_i$ is the number of secondary factor $x_i$ internal hazard points; and $S_i$ is the number of units containing the secondary factor $x_i$.

(2)  AHP (Analytic Hierarchy Process)

The analytic hierarchy process is an analytical method for calculating the relative importance of each evaluation factor by decomposition. The relative importance of each factor is scored from 1 to 9, from less important to more important. A consistency test is carried out later to calculate the random consistency ratio (CR) of the scoring scale. When

CR < 0.1, the consistency test is passed. According to the following formula, the landslide susceptibility index (LSI) of geological hazard can be calculated [24,25]:

$$LSI_{AHP} = w_1 \times AHP_{XI} + w_2 \times AHP_{X2} + \ldots \ldots + w_i \times AHP_{Xi} \tag{2}$$

In the formula, *AHP* represents the weight of each evaluation index subclass and *w* is the weight of each evaluation index.

(3)    Weighted Information Method

The information content method is used to calculate the second-level factors for each evaluation index, and the information values of the second-level factors for each evaluation index are obtained. Then, the analytic hierarchy process is used to weight and score the relative importance of each evaluation index and calculate the weight of each evaluation index. The geological hazard landslide susceptibility index (LSI) of the study area was calculated according to the following formula:

$$LSI = I_1 \times AHP_{XI} + I_2 \times AHP_{X2} + \ldots \ldots + I_i \times AHP_{Xi} \tag{3}$$

In the formula, *AHP* represents the weight of each evaluation index and *I* is the information value of the second-level factor for each evaluation index.

### 3.1.2. Establishment of the Risk Evaluation Method

Assessment of the predisposition to geological hazards only considers the static factors that affect the development of geological hazards and does not consider the predisposing factors that affect the formation of geological hazards in specific time periods. In this paper, by analyzing the formation mechanism of geological hazards in the study area, with rainfall as the main inducing factor for geological hazards in the study area, the geological hazard risk evaluation method based on an extremum hypothesis (i.e., based on the study area with the facts of the geological hazards in history, depending on access to the monitoring records, with maximum daily rainfall ($L_{max/day}$) as a hazard trigger) was employed, and the risk index for each evaluation unit (*H*) was calculated [26]:

$$H_i = \frac{Y_i}{Y_{max}} \cdot P_i \tag{4}$$

In the formula, $H_i$ is the risk index of the evaluation unit under a certain working condition; $Y_i$ is the susceptibility index of the *i*th evaluation unit; $Y_{max}$ is the maximum susceptibility index; and $P_i$ is the instability probability of the *I* evaluation unit in a given period under a certain working condition. $P_i = L/L_{max/day}$, $L_{max/day}$ is the daily maximum rainfall in the study area since the monitoring record began and *L* is the annual average maximum daily rainfall under different working conditions (heavy rain, rainstorm, heavy rainstorm, and extraordinary rainstorm).

The above formula input applications were carried out in ArcGIS software. The information values for each evaluation index in each unit of the study area were calculated, and the weighted information values were calculated by assigning the corresponding weights to each evaluation index. Finally, all evaluation indexes were superimposed to obtain the final susceptibility index for the study area. On this basis, the risk indexes for the study area evaluation unit were superimposed to obtain the final risk zoning results of the study area.

### 3.2. Selection of Evaluation Factors
### 3.2.1. Selection of Susceptibility Evaluation Factors

By identifying the development characteristics and formation mechanisms of the geological hazards of collapse and landslide in Yuanyang County and analyzing the geological environment of Yuanyang County, the evaluation indexes for susceptibility to geological hazards of collapse and landslide in Yuanyang County were divided into two levels: basic

factors and inducing factors. The basic factors were selected as the first-level evaluation indexes of four environmental factors: topography, geological structure, stratigraphic lithology, and water system [27]. Elevation, slope, and elevation differences play an important role in controlling the development of collapse and landslide. With increases in slope and elevation differences, the probabilities of collapse and landslide will also increase [28,29]. Different slope aspects have great influences on the illumination, precipitation and vegetation development of the slope, which affect the physical characteristics of rock and soil masses and thus the stability of slope blocks [30]. The internal and external forces reflected by different slope types also control the development of collapse and landslide [31]. Different landform types also have different controlling effects on the development of collapse and landslide [32]. Collapse and landslide hazards are closely related to geological structure. The complexity of the geological structure and the development of a fault zone in the study area affects the stability differences between slope bodies [33]. Stratigraphic lithology is an important internal factor and foundation that affects the development of collapse and landslide [34]. In order to make the evaluation results more scientific and accurate, the rock and soil masses in the study area were divided in detail according to their genesis, physical and mechanical strengths, lithology combinations, weathering degrees, and engineering geological characteristics. Firstly, soil bodies were classified according to soil property and structure. Secondly, rock masses were divided into five categories: a semi-diagenetic rock group, a magmatic rock group, a metamorphic clastic rock group, a sedimentary clastic rock group, and a carbonate rock group. Then, the engineering geological rock group was divided according to rock mass detail structure, karstification degree, weathering degree, rock strength, and lithology. The erosion and scouring of the river at its foot are the main factors affecting the instability of a bank slope. The shorter the distance from the river, the greater the probability of collapse and landslide [35]. The type of land use is also closely related to the occurrence of geological hazards. The regional geological hazards that tend to be associated with construction land and other human engineering activities are extraordinarily high [36]. Therefore, in order to analyze and evaluate susceptibility to collapse and landslide geological hazards in Yuanyang County more comprehensively and accurately, 9 factors, including engineering geological rock group, geomorphic types, distance from fault, distance from river, elevation, slope, aspect, curvature, and land-use type were selected as evaluation indicators, as shown in Figure 3.

In SPSS, the RF (random forest) model was used to analyze the quantitative relationship between the distribution of collapse and landslide hazard points and the classification of each evaluation factor, and the importance of each evaluation factor regarding susceptibility to collapse and landslide geological disasters in Yuanyang County was determined. An RF model is a machine-learning algorithm that trains and predicts samples through multiple decision trees [37]. Based on the original 228 collapse and landslide hazard points in the study area, the same number of non-collapse and non-landslide hazard points were randomly generated, and the corresponding evaluation factor attributes were extracted to form 456 total sample data sets for use in the RF model operation. Firstly, $m$ independent samples were randomly selected from the original data set $D$ using self-help sampling technology to generate $Dm$ training data sets. Secondly, a corresponding decision tree model $Tm$ was established for each training set, and $Rm$ classification results were obtained. Finally, the classification results were voted on, and the highest repetition was summarized as the optimal classification result of the model [34]. The RF model constructs different training sets by random sampling to generate decision trees, so it does not need to consider the interactions between factors and solves the problem of over-fitting. Finally, the importance of each evaluation factor in the classification model can be used to explain the impact on the development of collapse and landslide. It can be seen in Figure 4 that the abovementioned 9 evaluation factors selected have certain controlling effects on the related developments of landslide geological hazards, so the selected 9 evaluation factors could be used as geological hazard susceptibility evaluation factors in the calculations.

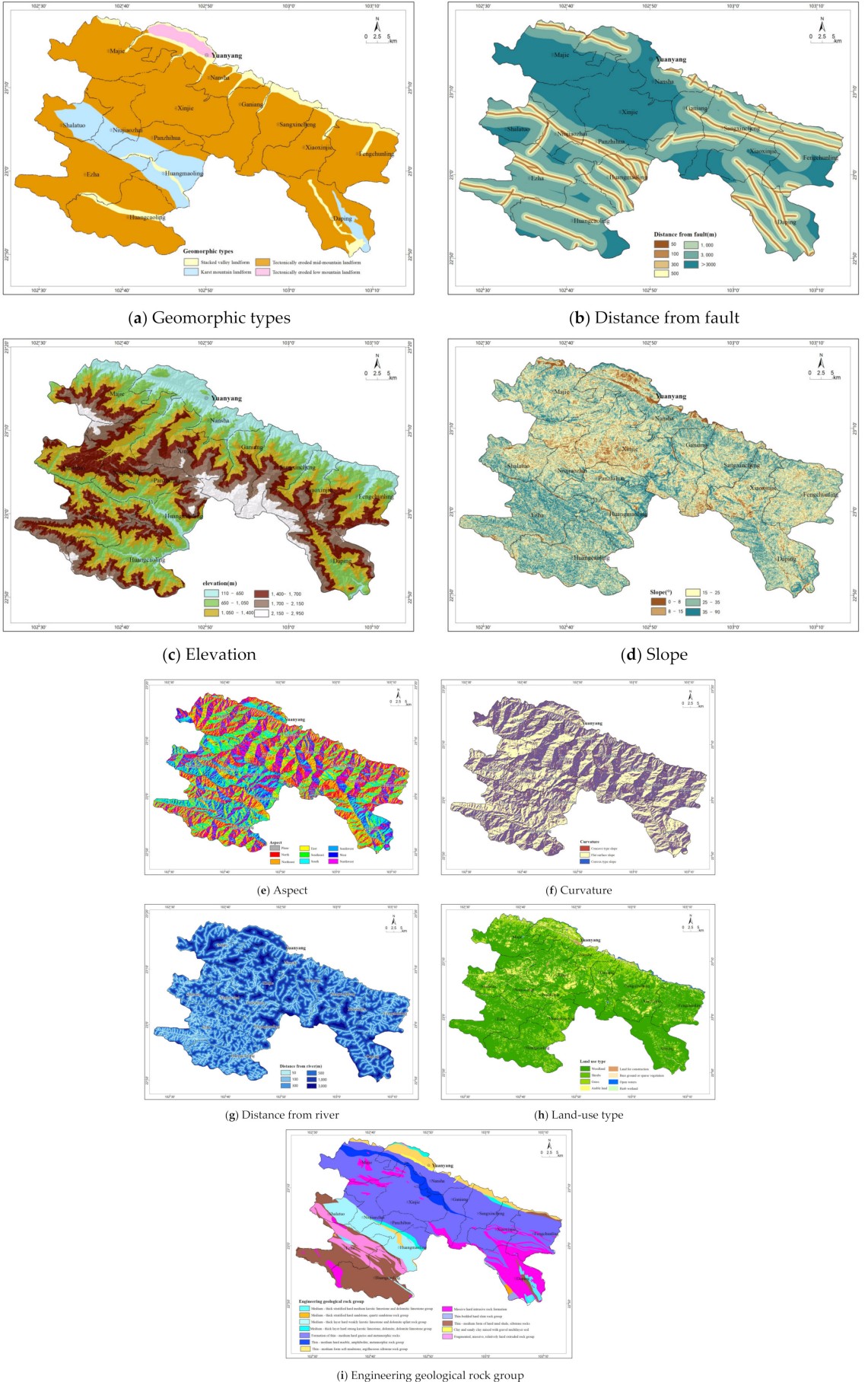

**Figure 3.** Evaluation factor grading diagram.

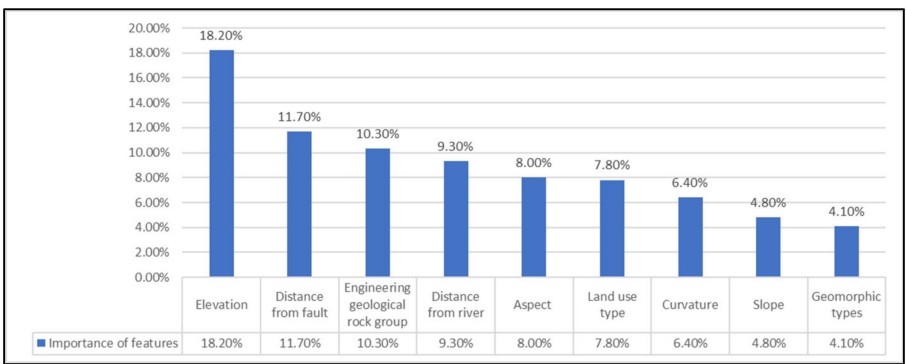

**Figure 4.** Feature importance of evaluation factors.

### 3.2.2. Selection of Risk Evaluation Factors

According to the data analysis and field investigation, it was found that the geological hazards of collapse and landslide in the study area were all related to rainfall, that the probabilities of geological hazards of collapse and landslide increased significantly during periods of rainfall and that, with increase in rainfall, the probability of geological hazards also increased. Therefore, the selection of rainfall parameters under different rainfall conditions is key to the accurate classification of geological hazard risk results for collapse and landslide under different rainfall conditions.

According to the provisions of the Flood Control Manual of the Chinese Control Office, rainfall is defined as the depth of the water layer that falls on the ground at a certain point or in a unit area within a certain time calculated in millimeters. Cumulative rainfall over 24 h of 25–50 mm is defined as heavy rain, 50–100 mm as rainstorm, 100–250 mm as heavy rainstorm, and more than 250 mm as extraordinary rainstorm.

Based on the annual average maximum daily rainfall (1948–2020) data obtained since detection records in the study area began, it can be seen from formula 4 that the hazard index for the evaluation unit under different rainfall conditions is obtained by combining the instability probability of the evaluation unit under different rainfall conditions. With the increase in rainfall level $L$, the slope instability probability $P_i$ of the evaluation unit increases, and the hazard index $H_i$ of the evaluation unit will increase accordingly, the probability of geological hazards also increasing. Therefore, the maximum annual average daily rainfall is taken as the inducing factor, and heavy rain, rainstorm, heavy rainstorm, and extraordinary rainstorm are divided into sections to carry out the risk evaluation of geological hazards of collapse and landslide under different rainfall conditions in the study area.

### *3.3. Evaluation Unit Demarcation*

(1)  Grid Units

Regular raster cells can process data faster and with higher accuracy and can synthesize each index. Therefore, this paper adopted the raster cell as one of the evaluation units. The following formula was used to calculate the optimal size of raster cells [38]:

$$G_f = 7.49 + 0.0006f - 2.0 \times 10^{-9}f^2 + 2.9 \times 10^{-15}f^3 \tag{5}$$

In the formula, $G_f$ is the appropriate mesh size and $f$ is the denominator value of contour accuracy. $f$ depends on the accuracy of the DEM (digital elevation model) for the study area.

The DEM data used in the study area were 1:50,000, so the suitable grid size could be calculated by the formula to be 30 m × 30 m; the study area was divided into 12,985,257 grid units (Figure 5).

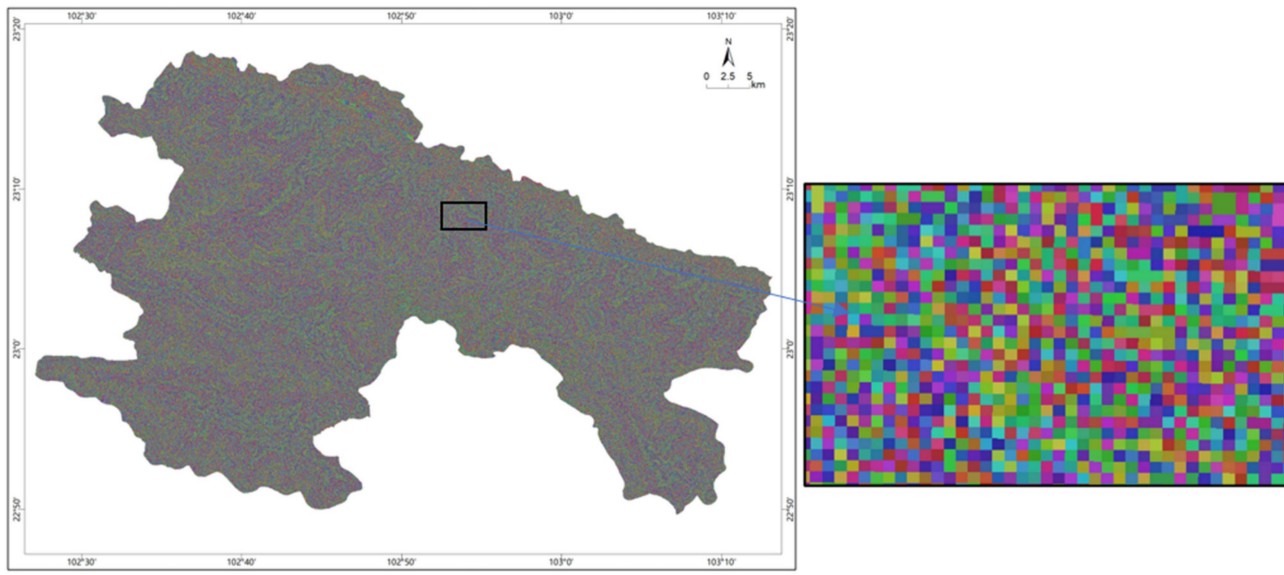

**Figure 5.** Grid unit division results.

(2)     Slope Units

A reasonable slope unit fully combines the topography of the study area and is suitable for study areas with complex geological structures. Therefore, the slope unit was used as one of the evaluation units in this paper. Slope unit division was performed via surface hydrological analysis based on a DEM [7,39–41]. The curvature was generated by a DEM, and a depression was identified according to the flow direction of a river. After inversion, the concave and convex geomorphic boundaries were identified to form the initial slope unit surface. Finally, the final slope unit division was obtained by manually trimming the unreasonable unit boundaries (Figure 6). A process operation in ArcGIS divided the study area into 7851 slope units (Figure 7).

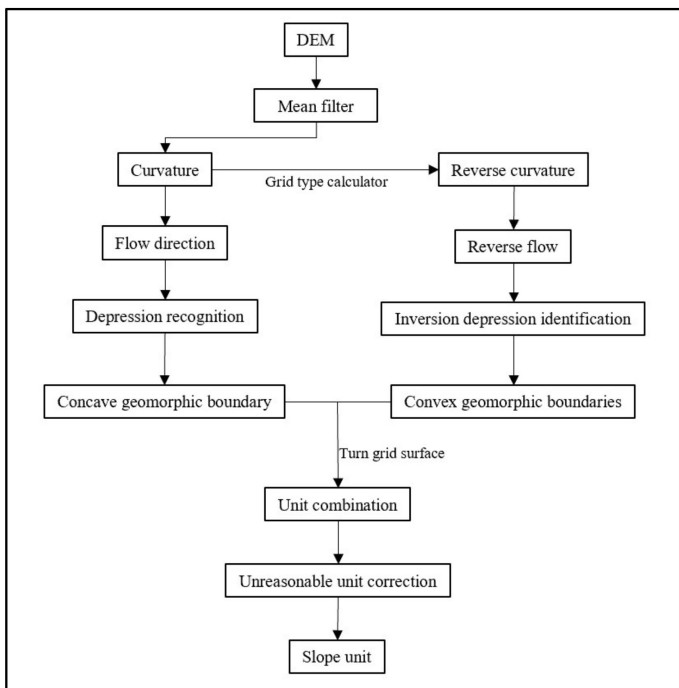

**Figure 6.** Flow chart of slope unit division.

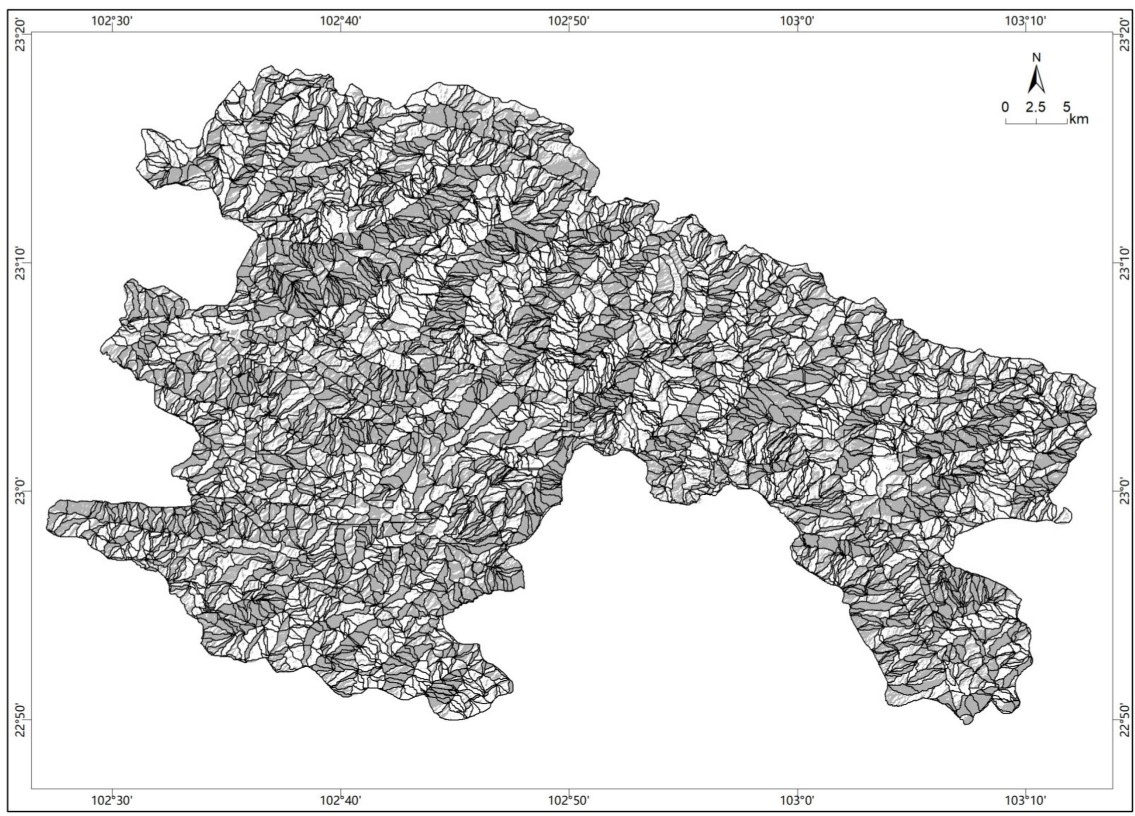

**Figure 7.** Result of slope unit division.

Grid unit division is simple and easy to calculate, but it is not practical for an area with large variations in terrain and a complex geological structure. As the basic unit of landslide development, the slope unit takes topographic terrain into account, but the differences between the geological environments in different regions are not considered in the same unit and the operation speed of the division calculation is slow. As the technology matures, the gap between the efficiency and speed of grid unit and slope unit divisions is narrowing.

*3.4. Regionalization of Collapse and Landslide Geological Hazard Susceptibilities*

Based on an objective statistical analysis of the data, in this study, the same standard was applied for grid units and slope units, and the influences of different evaluation factors regarding the susceptibility to collapse and landslide were calculated by the weighted information method. Although there were differences in individual variables, they were relatively consistent, and the differences in the statistical results for these individual variables were also normal.

Each evaluation factor of the grid unit was reclassified and given the corresponding weighted information value, and each slope unit was also given the weighted information value of each evaluation factor. Finally, the total weighted information value under the two evaluation units was calculated using the weighted information value formula. The information values of the second-level factors of the 9 evaluation indicators were calculated by the information content method, and the weight of each evaluation index was scored by the analytic hierarchy process. The calculated CR value was 0.0462, which was less than 0.1 and passed the consistency test (Table 1). Then, according to the weighted information value of the two evaluation units, it was divided into four prone areas, namely, low, medium, high and very high, according to the natural discontinuous point method, as shown in Figure 8.

**Table 1.** Comparison table of weighted information values.

| Evaluation Factors | Classification | $S_i$ | | $S_i/S$ | | $N_i$ | $N_i/N$ | Amount of Information | | Weight | CR |
|---|---|---|---|---|---|---|---|---|---|---|---|
| | | Grid Unit | Slope Unit | Grid Unit | Slope Unit | | | Grid Unit | Slope Unit | | |
| Geomorphic types | Stacked valley landform | 238,887 | 672 | 0.07 | 0.09 | 22 | 0.09 | 0.25 | 0.05 | | |
| | Karst mountain landform | 238,887 | 726 | 0.07 | 0.09 | 24 | 0.10 | 0.33 | 0.06 | 0.0391 | |
| | Tectonically eroded mid-mountain landform | 2,873,413 | 6446 | 0.84 | 0.82 | 199 | 0.81 | −0.04 | −0.01 | | |
| | Tectonically eroded low mountain landform | 53,971 | 52 | 0.02 | 0.01 | 0 | 0.00 | 0.00 | 0.00 | | |
| Land-use type | Woodland | 2,601,414 | 5757 | 0.73 | 0.73 | 120 | 0.49 | −0.41 | −0.40 | | |
| | Shrubs | 2993 | 7 | 0.00 | 0.00 | 0 | 0.00 | 0.00 | 0.00 | | |
| | Grass | 580,768 | 1095 | 0.16 | 0.14 | 19 | 0.08 | −0.75 | −0.58 | | |
| | Arable land | 232,178 | 562 | 0.07 | 0.07 | 7 | 0.03 | −0.83 | −0.91 | 0.0680 | |
| | Land for construction | 30,571 | 84 | 0.01 | 0.01 | 44 | 0.18 | 3.04 | 2.83 | | |
| | Bare ground or sparse vegetation | 78,435 | 226 | 0.02 | 0.03 | 55 | 0.22 | 2.32 | 2.06 | | |
| | Open waters | 14,991 | 73 | 0.00 | 0.01 | 0 | 0.00 | 0.00 | 0.00 | | |
| | Herb wetland | 61 | 2 | 0.00 | 0.00 | 0 | 0.00 | 0.00 | 0.00 | | |
| Distance from fault (m) | 50 | 63,229 | 156 | 0.02 | 0.02 | 4 | 0.02 | −0.09 | −0.19 | | |
| | 100 | 63,331 | 160 | 0.02 | 0.02 | 5 | 0.02 | 0.13 | 0.01 | | |
| | 300 | 246,527 | 581 | 0.07 | 0.07 | 28 | 0.11 | 0.50 | 0.44 | | |
| | 500 | 235,754 | 496 | 0.07 | 0.06 | 23 | 0.09 | 0.34 | 0.40 | 0.1314 | |
| | 1000 | 541,180 | 1158 | 0.15 | 0.15 | 38 | 0.16 | 0.01 | 0.06 | | |
| | 3000 | 1,281,573 | 2767 | 0.36 | 0.35 | 91 | 0.37 | 0.03 | 0.06 | | |
| | >3000 | 1,109,778 | 2578 | 0.31 | 0.33 | 56 | 0.23 | −0.32 | −0.36 | | |
| Elevation (m) | 110–650 | 408,649 | 913 | 0.12 | 0.12 | 21 | 0.09 | −0.30 | −0.30 | | |
| | 650–1050 | 729,924 | 1652 | 0.21 | 0.21 | 23 | 0.09 | −0.79 | −0.80 | | |
| | 1050–1400 | 846,548 | 1900 | 0.24 | 0.24 | 52 | 0.21 | −0.12 | −0.13 | 0.1634 | |
| | 1400–1700 | 811,256 | 1681 | 0.23 | 0.21 | 118 | 0.48 | 0.74 | 0.82 | | |
| | 1700–2150 | 545,512 | 1241 | 0.15 | 0.16 | 30 | 0.12 | −0.23 | −0.25 | | |
| | 2150–2950 | 199,476 | 509 | 0.06 | 0.06 | 1 | 0.00 | −2.62 | −2.76 | | |
| Engineering geological rock group | Thin–medium-form soft mudstone, argillaceous siltstone rock group | 142,964 | 205 | 0.04 | 0.03 | 3 | 0.01 | −1.19 | −0.75 | | |
| | Medium–thick layer of hard strong karstic limestone, dolomite, dolomite limestone group | 55,908 | 93 | 0.02 | 0.01 | 1 | 0.00 | −1.35 | −1.06 | | |
| | Clay and sandy clay mixed with gravel multilayer soil | 14,812 | 59 | 0.00 | 0.01 | 1 | 0.00 | −0.02 | −0.60 | | |
| | Formation of thin–medium hard gneiss and metamorphic rocks | 1,658,434 | 3880 | 0.47 | 0.49 | 117 | 0.48 | 0.02 | −0.03 | | |
| | Thin–medium hard marble, amphibolite, metamorphic rock group | 131,907 | 327 | 0.04 | 0.04 | 4 | 0.02 | −0.82 | −0.93 | 0.0462 | |
| | Thin–medium form of hard mud shale, siltstone rocks | 541,466 | 1198 | 0.15 | 0.15 | 49 | 0.20 | 0.27 | 0.28 | 0.0888 | |
| | Massive hard intrusive rock formation | 441,545 | 1011 | 0.12 | 0.13 | 38 | 0.16 | 0.22 | 0.19 | | |
| | Fragmented, massive, relatively hard extruded rock group | 196,231 | 398 | 0.06 | 0.05 | 7 | 0.03 | −0.66 | −0.57 | | |
| | Medium–thick layer of hard weakly karstic limestone and dolomite splint rock group | 303,885 | 613 | 0.09 | 0.08 | 23 | 0.09 | 0.09 | 0.19 | | |
| | Medium–thick stratified hard sandstone, quartz sandstone rock group | 7250 | 18 | 0.00 | 0.00 | 2 | 0.01 | 1.38 | 1.28 | | |
| | Medium–thick stratified hard medium karstic limestone and dolomitic limestone group | 29,680 | 67 | 0.01 | 0.01 | 0 | 0.00 | 0.00 | 0.00 | | |
| | Thin-bedded hard slate rock group | 16,893 | 27 | 0.00 | 0.00 | 0 | 0.00 | 0.00 | 0.00 | | |
| Slope (°) | 0–8 | 131,433 | 1022 | 0.04 | 0.13 | 7 | 0.03 | −0.27 | −1.51 | | |
| | 8–15 | 450,167 | 1374 | 0.13 | 0.17 | 51 | 0.21 | 0.49 | 0.18 | | |
| | 15–25 | 1,235,738 | 2487 | 0.35 | 0.31 | 103 | 0.42 | 0.18 | 0.29 | 0.1694 | |
| | 25–35 | 1,161,406 | 2022 | 0.33 | 0.26 | 64 | 0.26 | −0.23 | 0.02 | | |
| | 35–90 | 550,193 | 991 | 0.16 | 0.13 | 20 | 0.08 | −0.65 | −0.43 | | |
| Aspect | Plane | 28,351 | 79 | 0.01 | 0.01 | 1 | 0.00 | −0.68 | −0.90 | | |
| | North | 315,913 | 636 | 0.09 | 0.08 | 19 | 0.08 | −0.14 | −0.04 | | |
| | Northeast | 534,362 | 1362 | 0.15 | 0.17 | 37 | 0.15 | 0.00 | −0.13 | | |
| | East | 405,228 | 916 | 0.11 | 0.12 | 31 | 0.13 | 0.10 | 0.09 | | |
| | Southeast | 449,606 | 981 | 0.13 | 0.12 | 29 | 0.12 | −0.07 | −0.05 | 0.0870 | |
| | South | 408,220 | 859 | 0.12 | 0.11 | 38 | 0.16 | 0.30 | 0.35 | | |
| | Southwest | 335,603 | 804 | 0.09 | 0.10 | 21 | 0.09 | −0.10 | −0.17 | | |
| | West | 317,065 | 697 | 0.09 | 0.09 | 22 | 0.09 | 0.00 | 0.02 | | |
| | Northwest | 442,634 | 958 | 0.13 | 0.12 | 27 | 0.11 | −0.13 | −0.10 | | |
| | North | 298,057 | 604 | 0.08 | 0.08 | 20 | 0.08 | −0.03 | 0.07 | | |
| Curvature | Concave type slope | 934,362 | 2222 | 0.26 | 0.28 | 49 | 0.20 | −0.28 | −0.34 | | |
| | Flat surface slope | 1,528,094 | 3333 | 0.43 | 0.42 | 121 | 0.49 | 0.13 | 0.16 | 0.0304 | |
| | Convex type slope | 1,069,874 | 2341 | 0.30 | 0.30 | 75 | 0.31 | 0.01 | 0.03 | | |

**Table 1.** *Cont.*

| Evaluation Factors | Classification | $S_i$ | | $S_i/S$ | | $N_i$ | $N_i/N$ | Amount of Information | | Weight | CR |
|---|---|---|---|---|---|---|---|---|---|---|---|
| | | Grid Unit | Slope Unit | Grid Unit | Slope Unit | | | Grid Unit | Slope Unit | | |
| Distance from river (m) | 50 | 419,847 | 1579 | 0.12 | 0.20 | 33 | 0.13 | 0.13 | −0.40 | | |
| | 100 | 376,932 | 628 | 0.11 | 0.08 | 25 | 0.10 | −0.04 | 0.25 | | |
| | 300 | 1,189,564 | 2375 | 0.34 | 0.30 | 74 | 0.30 | −0.11 | 0.00 | 0.0532 | |
| | 500 | 746,745 | 1472 | 0.21 | 0.19 | 55 | 0.22 | 0.06 | 0.19 | | |
| | 1000 | 698,574 | 1562 | 0.20 | 0.20 | 55 | 0.22 | 0.13 | 0.13 | | |
| | 3000 | 109,676 | 280 | 0.03 | 0.04 | 3 | 0.01 | −0.93 | −1.06 | | |

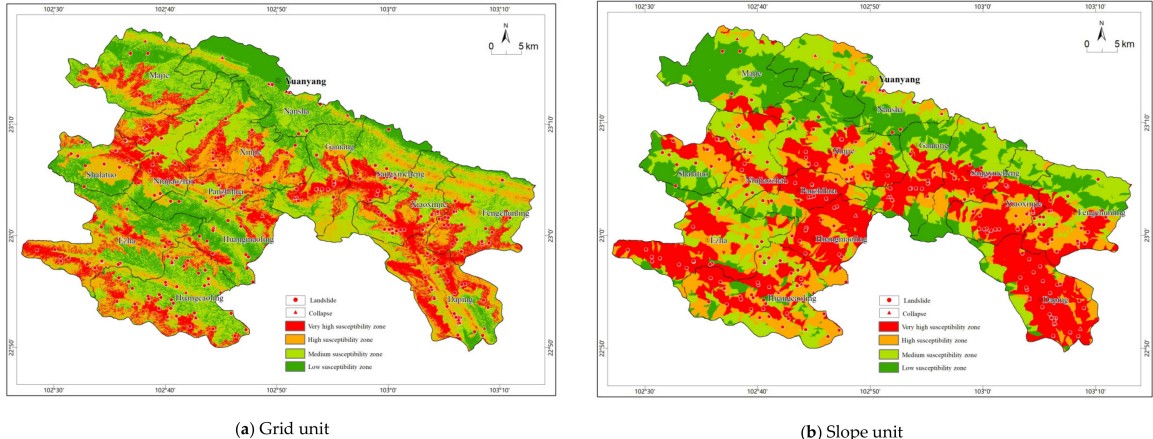

(**a**) Grid unit      (**b**) Slope unit

**Figure 8.** Geological hazard susceptibility zoning of collapse and landslide.

### 3.5. Regionalization of Geological Hazard Risk of Collapse and Landslide

In this paper, according to the regulations of the National Prevention Office of China, the annual average maximum daily rainfall (1948–2020) in the study area was selected as the basis. The geological hazard risk of collapse and landslide in the study area was divided into geological hazard risk divisions under heavy rain (25–50 mm), rainstorm (50–100 mm), heavy rainstorm (100–250 mm), and extraordinary rainstorm (>250 mm) as four types of risk zoning. On the basis of the evaluation of the susceptibility to geological hazards in the study area, the instability probabilities of the evaluation units under different rainfall conditions were superimposed by ArcGIS, and the risk indexes for the slope units under different conditions were obtained, these being divided into four grades of very high risk area, high-risk area, medium-risk area, and low-risk area, according to the natural discontinuous point method (Figures 9–12).

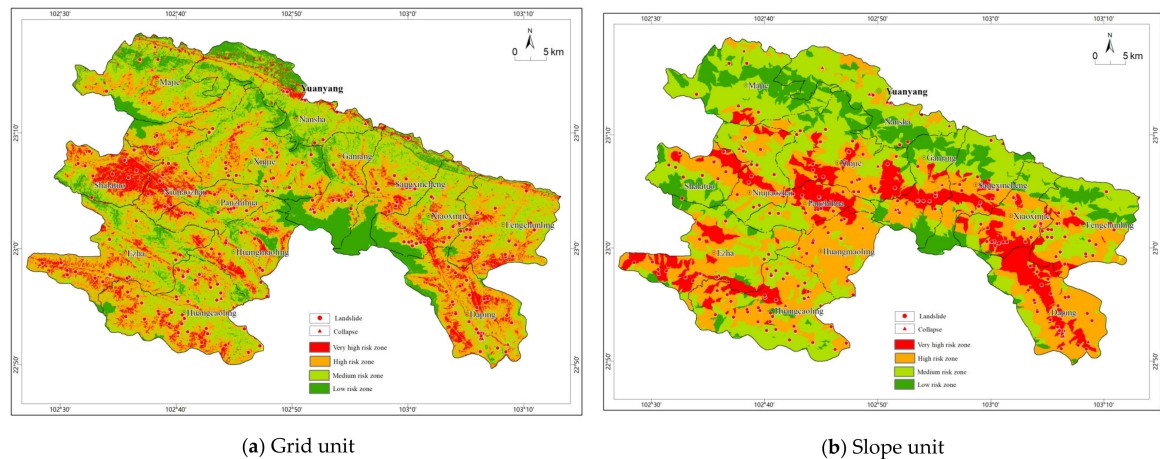

(**a**) Grid unit      (**b**) Slope unit

**Figure 9.** Geological risk zoning map of collapse and landslide under heavy rain conditions.

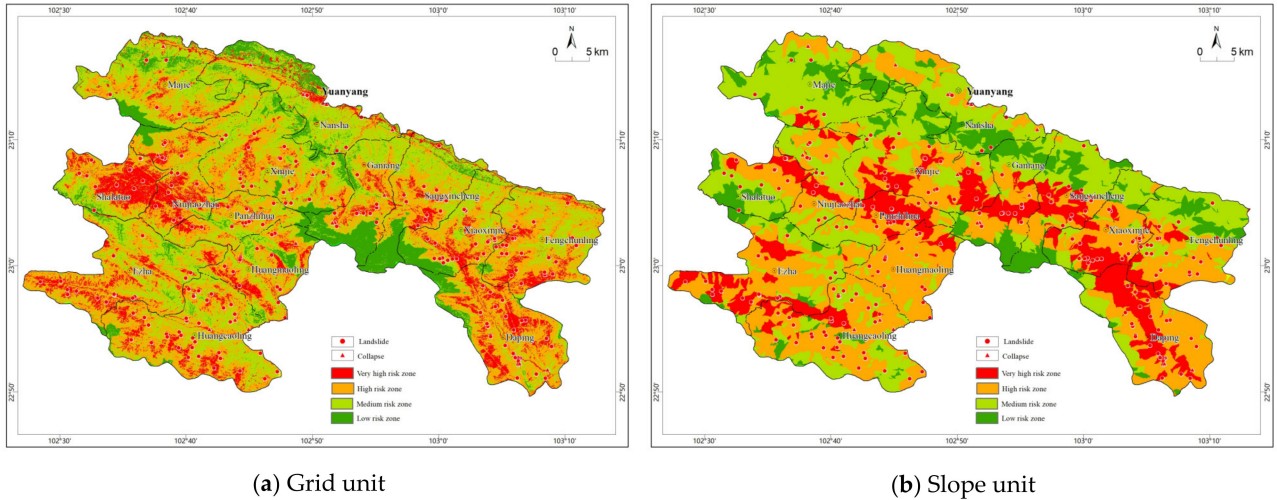

(**a**) Grid unit                      (**b**) Slope unit

**Figure 10.** Geological risk zoning map of collapse and landslide under rainstorm conditions.

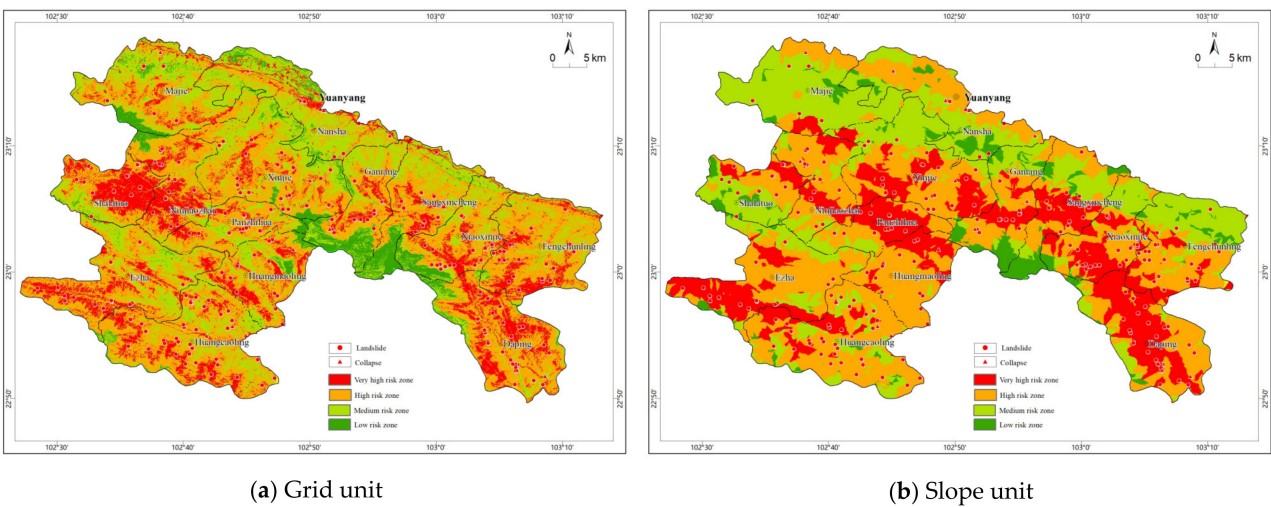

(**a**) Grid unit                      (**b**) Slope unit

**Figure 11.** Geological risk zoning map of collapse and landslide under heavy rainstorm conditions.

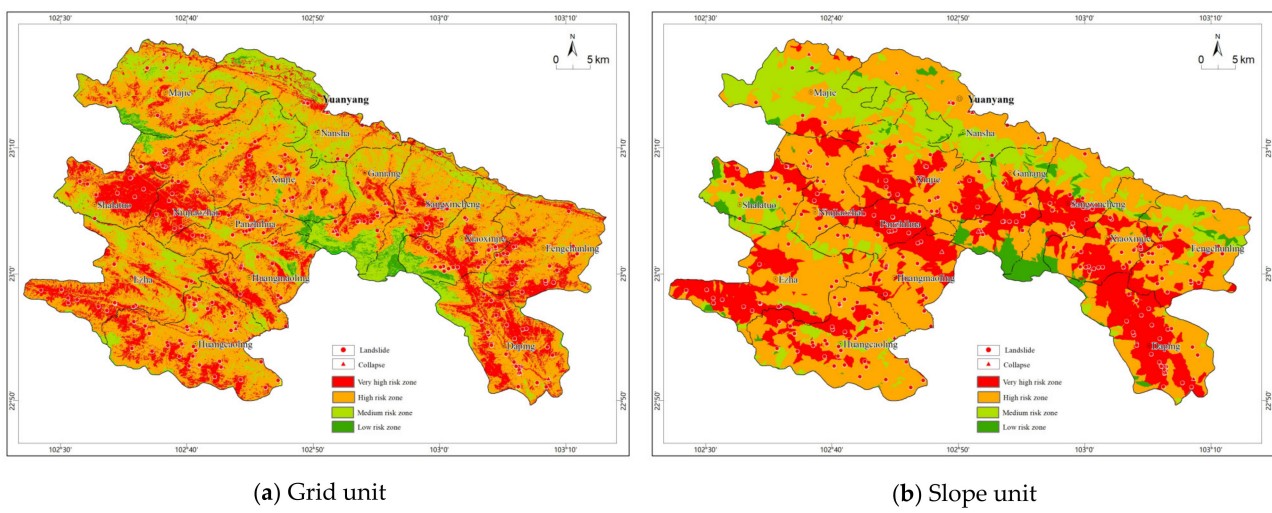

(**a**) Grid unit                      (**b**) Slope unit

**Figure 12.** Geological risk zoning map of collapse and landslide under extraordinary rainstorm conditions.

## 4. Comparison of Results

### *4.1. Result Data Comparison*

4.1.1. Comparison of Susceptibility Result Data

This paper takes the number of hazard points contained in the very high and high-susceptibility areas as the basis for comparison. Under different evaluation units, the very high and high-susceptibility areas contained more hazard points, and the evaluation results were more reasonable when it came to judging the accuracy of the evaluation results for the two evaluation units. With an increase in the susceptibility level, the hazard density for the two evaluation units increased, showing a good positive correlation (Figure 13), which conforms to the classification principle regarding the susceptibility level of geological hazards. The proportion of hazards in very high susceptibility areas was the highest and twice that in high-susceptibility areas. Comparing the hazard ratios for the slope units and the grid units, the slope unit values in the very high and high-susceptibility areas were 20.08% higher than those of the grid units, and the slope unit values in the low- and medium-susceptibility areas were 20.07% lower (Table 2). Therefore, the evaluation results for the slope units were more reasonable.

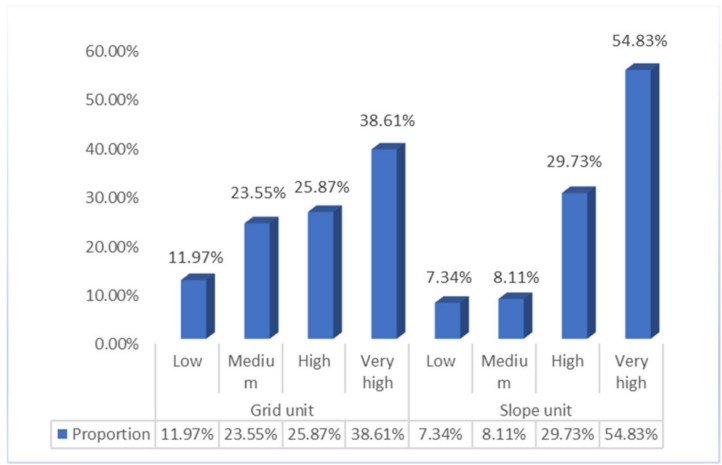

**Figure 13.** Statistical chart of the proportion of hazards resulting from susceptibility.

**Table 2.** Comparison table of susceptibility grades and hazard distributions.

| Evaluation Unit | Susceptibility Grade | Number of Damage Points | Proportion | Interval Area (km$^2$) | Proportion |
|---|---|---|---|---|---|
| Grid unit | Low | 31 | 11.97% | 475.55 | 21.55% |
| | Medium | 61 | 23.55% | 782.7 | 35.46% |
| | High | 67 | 25.87% | 556.89 | 25.23% |
| | Very high | 100 | 38.61% | 392.01 | 17.16% |
| Slope unit | Low | 19 | 7.34% | 410.06 | 18.58% |
| | Medium | 21 | 8.11% | 470.21 | 21.30% |
| | High | 77 | 29.73% | 550.02 | 24.92% |
| | Very high | 142 | 54.83% | 776.87 | 35.20% |

4.1.2. Comparison of Risk Result Data

Based on a comparison of the number of hazard points contained in the very high and high-risk areas, under different evaluation units, the very high and high-risk areas were found to contain more hazard points, and the evaluation results were more reasonable when it came to judging the accuracy of the evaluation results for the two evaluation units. In the evaluation results for geological hazard risk of collapse and landslide under four different rainfall conditions, the hazard density of the two evaluation units increased with the increase in risk level, showing a good positive correlation (Figure 14), which is in line with the classification principle of geological hazard risk level. The highest proportion of

hazards was found in the very high risk area, followed by the high-risk area. Comparing the hazard ratios for slope units and grid units, slope unit values in the very high and high-risk area were higher than the grid unit values, and slope unit values in the low- and medium-risk areas were lower than the grid unit values (Table 3). Therefore, the evaluation results for the slope units were more reasonable.

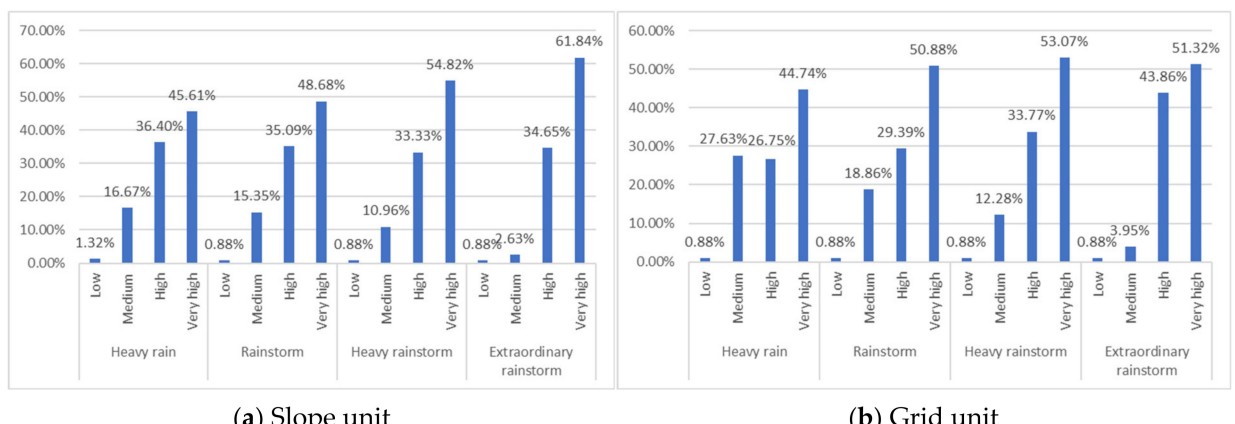

(**a**) Slope unit         (**b**) Grid unit

**Figure 14.** Statistical chart of risk result hazard proportions.

**Table 3.** Comparison table of risk grades and hazard distributions under different rainfall conditions.

| Evaluation Unit | Working Condition of Rainfall | Level of Risk | Area (km²) | Proportion (%) | Number of Damage Points | Proportion (%) |
|---|---|---|---|---|---|---|
| Slope unit | Heavy rain | Low | 285.47 | 12.88% | 3 | 1.32% |
| | | Medium | 897.58 | 40.51% | 38 | 16.67% |
| | | High | 796.35 | 35.94% | 83 | 36.40% |
| | | Very high | 236.47 | 10.67% | 104 | 45.61% |
| | Rainstorm | Low | 211.76 | 9.56% | 2 | 0.88% |
| | | Medium | 785.26 | 35.44% | 35 | 15.35% |
| | | High | 923.09 | 41.66% | 80 | 35.09% |
| | | Very high | 295.77 | 13.35% | 111 | 48.68% |
| | Heavy rainstorm | Low | 117.02 | 5.28% | 2 | 0.88% |
| | | Medium | 640.59 | 28.91% | 25 | 10.96% |
| | | High | 1077.75 | 48.64% | 76 | 33.33% |
| | | Very high | 380.52 | 17.17% | 125 | 54.82% |
| | Extraordinary rainstorm | Low | 46.52 | 2.10% | 2 | 0.88% |
| | | Medium | 391.71 | 17.68% | 6 | 2.63% |
| | | High | 1300.44 | 58.69% | 79 | 34.65% |
| | | Very high | 477.21 | 21.54% | 141 | 61.84% |
| Grid unit | Heavy rain | Low | 300.75 | 13.57% | 2 | 0.88% |
| | | Medium | 872.18 | 39.36% | 63 | 27.63% |
| | | High | 736.89 | 33.25% | 61 | 26.75% |
| | | Very high | 306.06 | 13.81% | 102 | 44.74% |
| | Rainstorm | Low | 248.97 | 11.24% | 2 | 0.88% |
| | | Medium | 765.52 | 34.55% | 43 | 18.86% |
| | | High | 860.20 | 38.82% | 67 | 29.39% |
| | | Very high | 341.19 | 15.40% | 116 | 50.88% |
| | Heavy rainstorm | Low | 118.80 | 5.36% | 2 | 0.88% |
| | | Medium | 644.32 | 29.08% | 28 | 12.28% |
| | | High | 954.51 | 43.08% | 77 | 33.77% |
| | | Very high | 498.24 | 22.49% | 121 | 53.07% |
| | Extraordinary rainstorm | Low | 79.85 | 3.60% | 2 | 0.88% |
| | | Medium | 356.23 | 16.08% | 9 | 3.95% |
| | | High | 1196.48 | 54.00% | 100 | 43.86% |
| | | Very high | 583.32 | 26.32% | 117 | 51.32% |

*4.2. Comparison of Model Accuracy*

In this paper, the ROC (receiver operating characteristic) curve method—a quantitative research method—was used as the optimal evaluation model. The areas under the ROC curves (AUCs) were used to judge prediction accuracy. The same number of hazard points and non-hazard points were randomly selected from the results of susceptibility and risk zoning under the grid unit and slope unit systems. The ROC curves were plotted based on the susceptibility and risk zoning level areas where the hazard points and non-hazard points were located, and the area under the curve (ACU) values were determined. At the same time, the performance ranking mechanism was used as the classification standard of prediction accuracy: AUC between 0.5 and 1: (1) fairly good: AUC greater than 0.9 and less than 1.0; (2) very good: AUC greater than 0.8 and less than 0.9; (3) good: AUC greater than 0.7 and less than 0.8; (4) poor: AUC greater than 0.6 and less than 0.7; (5) range: AUC greater than 0.5 and less than 0.6 [14,35,36,42–51].

### 4.2.1. Precision Comparison of Susceptibility Zoning Results

The ROC curves and AUC numerical results for the susceptibility zoning results under the slope unit and grid unit systems are shown in Figure 15. The AUC value based on grid unit susceptibility regionalization was 81.7%, falling into the "very good" region. The AUC value based on slope unit susceptibility regionalization was 92.7%, falling into the "fairly good" region. From the perspective of AUC values, the evaluation result based on the slope unit system was higher than that based on the grid unit system. From the ROC curve evaluation results, the slope unit system was more accurate than the grid unit system.

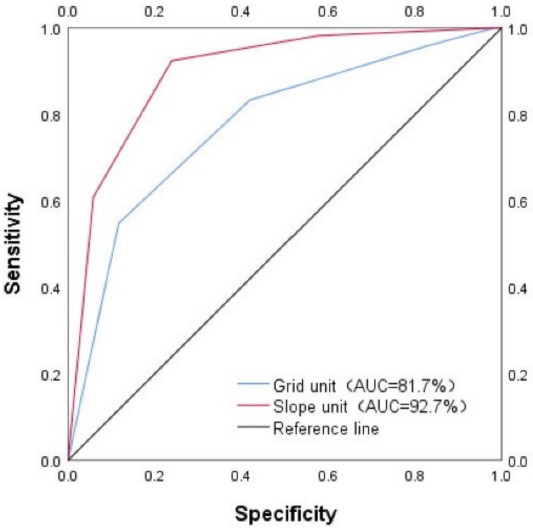

**Figure 15.** ROC curve comparison of susceptibility evaluation of different evaluation units.

### 4.2.2. Precision Comparison of Risk Zoning Results

Slope units and grid units under four different rainfall conditions under the condition of risk regionalization results for ROC curves compared with AUC numerical results, as shown in Figure 16; the risk evaluation results based on the slope unit evaluation system all fell into the "good" and above prediction accuracy area under the conditions of heavy rain, rainstorm, heavy rainstorm, and extraordinary rainstorm, and the prediction accuracy was better than the evaluation results under the grid unit evaluation system to a certain extent. From the perspective of AUC values, the evaluation results based on the slope unit system were higher than those based on the grid unit system. From the ROC curve evaluation results, the slope unit risk evaluation results were more accurate than the grid unit results.

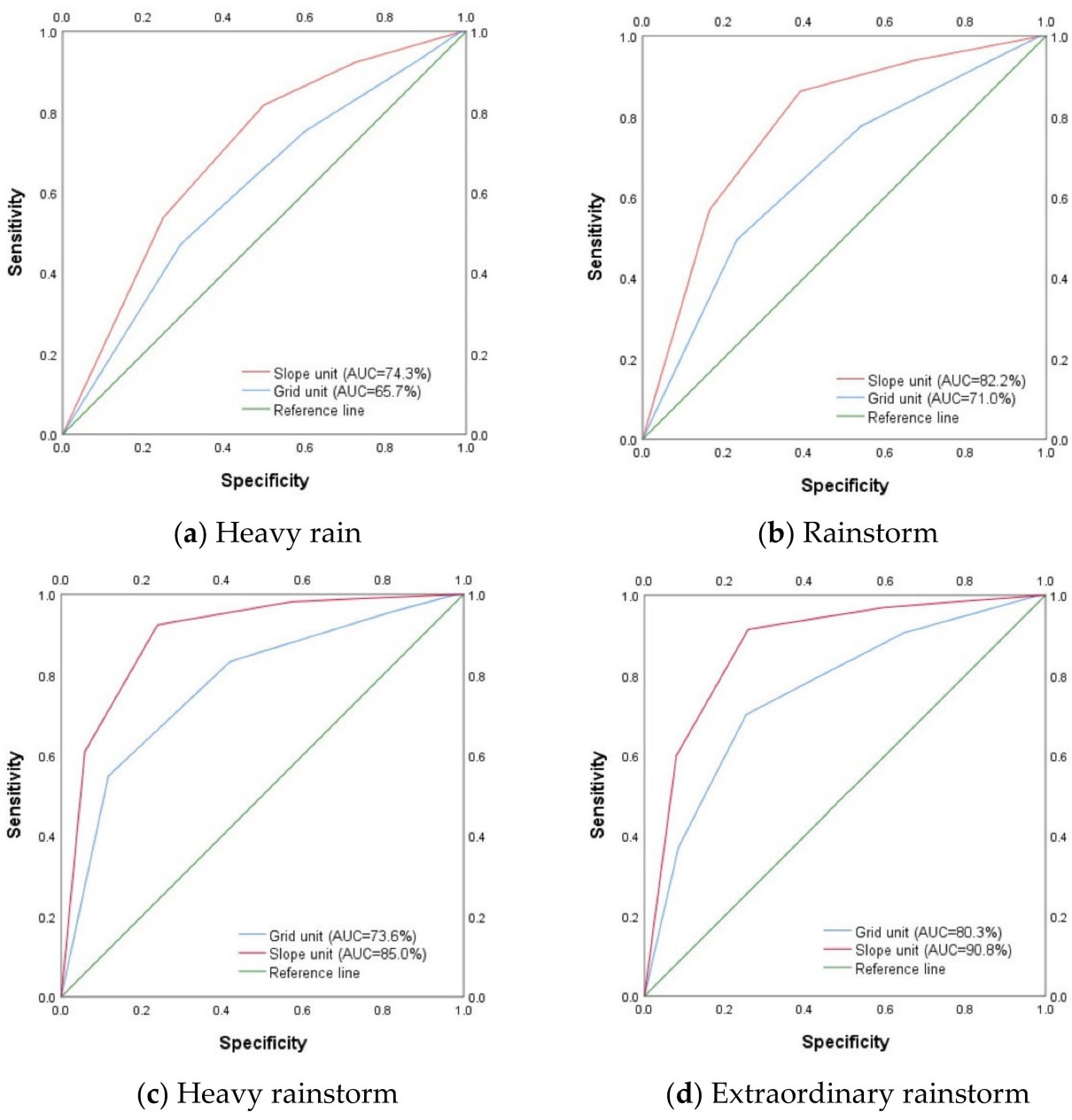

(**a**) Heavy rain

(**b**) Rainstorm

(**c**) Heavy rainstorm

(**d**) Extraordinary rainstorm

**Figure 16.** ROC curve comparison of risk evaluations of different rainfall conditions under different evaluation units.

*4.3. Comparison of Slope Unit Evaluation Results with the Actual Situation*

4.3.1. Results of the Susceptibility Evaluation

In this paper, the geological hazard susceptibility zoning results under the slope unit evaluation system with better evaluation results were compared with detailed geological hazard investigation results for Yuanyang County. By comparing the results of the susceptibility regionalization of the slope unit system in this paper with the results of the detailed investigation, it was found (Table 4, Figure 17) that the number of hazard points and the area occupied by the susceptibility regionalization results with the slope unit as the evaluation unit were consistent with the actual situation and that the regionalization results were relatively accurate.

**Table 4.** Comparison table of the results of the susceptibility zoning.

| Result Source | Susceptibility Grade | Number of Hazard Sites | Proportion | Interval Area (km²) | Proportion |
|---|---|---|---|---|---|
| Evaluation results of slope unit system | Low | 19 | 7.34% | 410.06 | 18.58% |
| | Medium | 21 | 8.11% | 470.21 | 21.30% |
| | High | 77 | 29.73% | 550.02 | 24.92% |
| | Very high | 142 | 54.83% | 776.87 | 35.20% |
| Evaluation results of detailed investigation | Low | 0 | 0.00% | 369.70 | 16.75% |
| | Medium | 19 | 7.34% | 511.18 | 23.16% |
| | High | 80 | 30.89% | 591.30 | 26.79% |
| | Very high | 157 | 60.62% | 734.98 | 33.30% |

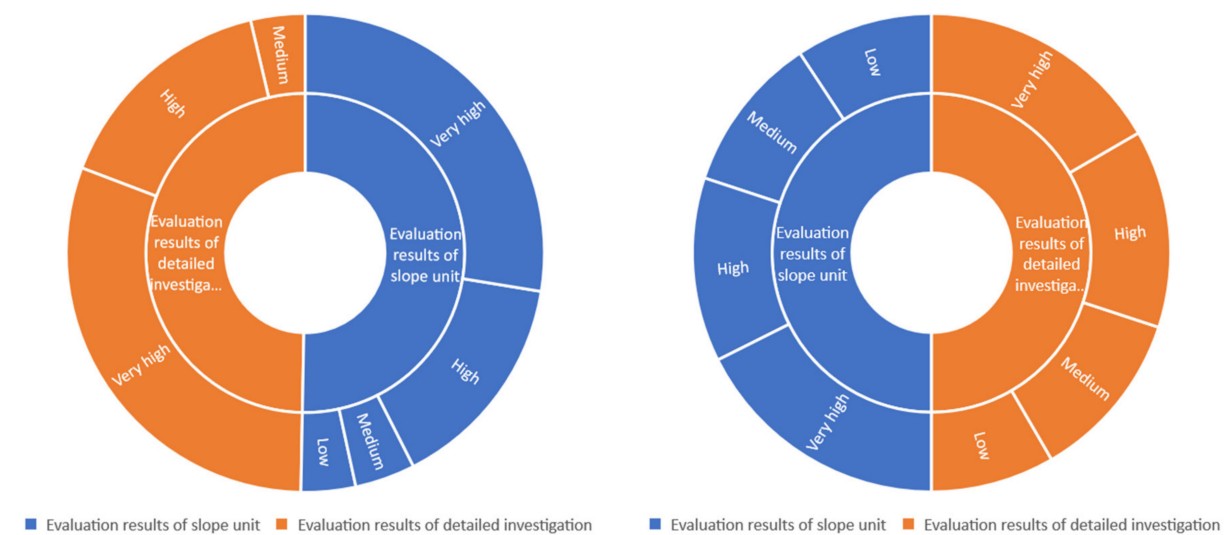

(**a**) Hazard points contained in the susceptibility zone   (**b**) Area occupied by the susceptibility zone

**Figure 17.** Comparison of the results for the susceptibility regionalization with the actual situation.

4.3.2. Results of Risk Evaluation

Similarly, the geological hazard risk zoning results under the slope unit evaluation system with better evaluation results were compared with the detailed geological hazard investigation results for Yuanyang County. Taking the results of the geological hazard risk zoning under different rainfall conditions, the maximum possible rainfall conditions in the study area were selected, and the results of geological hazard risk zoning under heavy rainstorm conditions were selected for data comparison with the results of actual detailed investigation of risk zoning in the study area.

By comparison, it was found that the area of the geological hazard risk zoning results under the condition of heavy rainstorm for the slope unit evaluation system was highly consistent with the actual detailed investigation results for the study area and the hazard numbers included (Table 5, Figure 18). The geological hazard risk assessment of collapse and landslide in the study area under the condition of heavy rainstorm had high accuracy. Since the risk evaluation results under the heavy rainstorm condition were obtained via the same evaluation system as those obtained under the other three conditions, the risk evaluation results for geological hazards of collapse and landslide in the study area under different rainfall conditions under the slope unit evaluation system were found to have good rationality and accuracy.

**Table 5.** Comparison of risk zoning results.

| Results the Source | Level of Risk | Interval Area (km²) | Proportion (%) | Number of Hazard Sites | Proportion (%) |
|---|---|---|---|---|---|
| Results for heavy rainstorm conditions (slope unit system) | Low | 117.02 | 5.28% | 2 | 0.88% |
| | Medium | 640.59 | 28.91% | 25 | 10.96% |
| | High | 1077.75 | 48.64% | 76 | 33.33% |
| | Very high | 380.52 | 17.17% | 125 | 54.82% |
| Results of a detailed survey of zoning in the study area | Low | 202.77 | 9.15% | 0 | 0.00% |
| | Medium | 603.41 | 27.23% | 17 | 7.46% |
| | High | 1035.42 | 46.73% | 81 | 35.53% |
| | Very high | 374.28 | 16.89% | 130 | 57.02% |

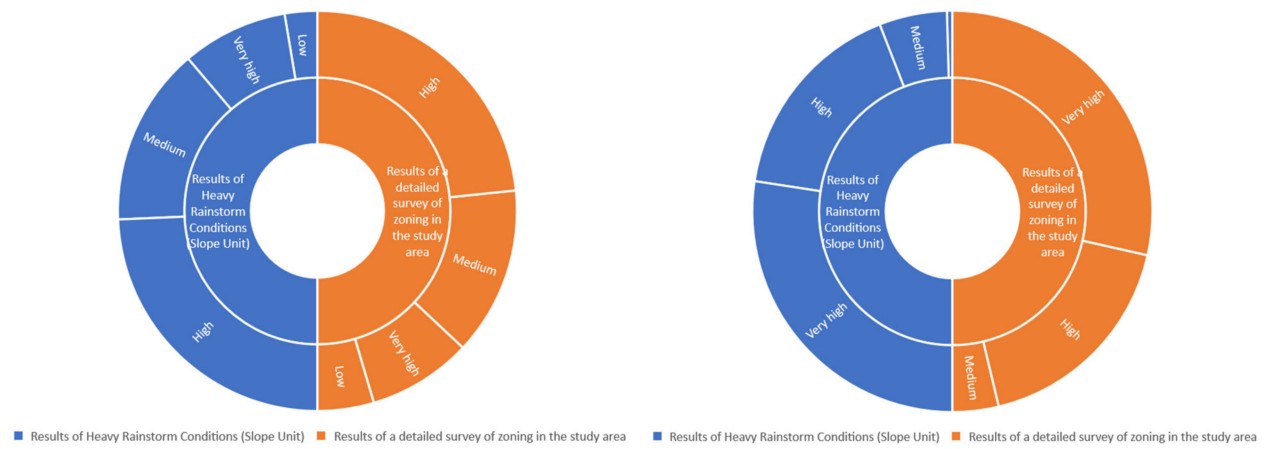

(**a**) Area occupied by the risk zone      (**b**) Hazard points contained in the risk zone

**Figure 18.** Comparison of the results for risk regionalization with the actual situation.

## 5. Discussion

Some scholars have used the RF model to evaluate susceptibility to geological hazards, and surface faults and rivers have the most important influences on the development of landslides [37,52,53]. This study also confirmed this point. Through the use of statistics and analysis of the distribution of hazard points and the evaluation factors, it was concluded that the distance between faults and rivers has a significant impact on landslides. The closer an area is to a river, the greater the water content of the soil, coupled with the undercutting effect of the surface runoff of rainfall on rock and soil mass, and the closer the area is to a fault, the stronger the geological structural activity, which promotes the development of collapse and landslide. In general, correlation between evaluation factors will bring informational redundancy to the prediction results of the model. As the information content model was used in this study to extract information after grading the evaluation factors, interaction between the factors was not considered.

The type of evaluation unit selected affects the accuracy of evaluation results [54–61]. In this paper, the evaluation results of grid unit and slope unit systems have been compared. The suitability of other evaluation units for geological hazard evaluation and the division of more units suitable for geological hazard evaluation need to be further studied and discussed. The size of the research area division unit also affects the accuracy and rationality of the evaluation results. The division method of the evaluation unit and the suitable size of the research area division unit also need further research and discussion.

Through comparison of the research results, it was found that there was a certain difference between the susceptibility level and the distribution of hazard points in the evaluation results based on the weighted information amount. This may be because the weighted information amount model does not take into account the influence of non-hazard points on the prediction of collapse and landslide susceptibility results. The susceptibility

evaluation model established on this basis cannot better reflect the impact mechanism of collapse and landslide [61–69]. In the future, hazard points and non-hazard points could be considered at the same time in a prediction model to evaluate susceptibility to geological hazards. Different study area data could also be selected to verify the accuracy of model prediction, enhance the scientific applicability and generalization of the model, and improve the prediction results of the evaluation model.

For the risk assessment of geological hazards, a large number of studies have mostly involved rainfall as an evaluation factor in model prediction [70]. Addressing this problem, this paper selected the annual average maximum daily rainfall in the study area and obtained hazard assessment results under four rainfall conditions: heavy rain, rainstorm, heavy rainstorm, and extraordinary rainstorm. Compared with the comprehensive index method used by Ji Y et al., based on the assumption of extreme rainfall, the annual maximum daily rainfall at different frequencies was used as the inducing factor and the slope unit was used as the evaluation unit to obtain results for geological disaster risk zoning under different rainfall conditions [26]. The advantage of this paper is that the evaluation results were found to be more accurate and reasonable using a combination of subjective weighting AHP and objective statistical ICM. In the selection of rainfall inducing factors, the classification of rainfall grade was based on the Chinese Flood Control Manual. In order to reasonably be used for predictions with geological hazard risk assessment models, accurate rainfall data and rainfall classifications of different rainfall conditions need to be collected and studied.

## 6. Conclusions

(1) In the study area, a total of 7851 slope units were divided by ArcGis using DEM data. The boundaries of slope units were highly consistent with the ridge lines and valley lines, and the divided slope units were in line with the topographic and geomorphic characteristics of the study area, indicating that the method proposed in this paper can be used to divide slope units in large study areas.

(2) Taking grid units and slope units as evaluation units and taking maximum annual average daily rainfall as the inducing factor, the risk evaluation results for collapse and landslide under four different rainfall conditions—heavy rain, rainstorm, heavy rainstorm, and extraordinary rainstorm—were obtained, which improved the disadvantage of using the rainfall inducing factor as a single evaluation factor in the calculations of the geological hazard risk evaluation model. The degrees and regional distribution characteristics of geological hazards induced by rainfall grade were clarified, providing a basis for the implementation of geological hazard prevention and control by means of prevention, avoidance, control, rescue, or a combination thereof, which is conducive to improving the operability and timeliness of hazard prevention and mitigation. At the same time, the geological hazard risk assessment system under different rainfall conditions proposed in this paper provides a reference for geological hazard risk assessment in other regions.

(3) In this paper, grid units and slope units were used as the evaluation units in a collapse and landslide geological hazard evaluation system. According to the statistical analysis and comparison of the results, the proportions of hazards in the very high and high-susceptibility areas and risk areas under the slope unit system were higher than those under the grid unit system. In the comparison of model accuracy tests, the AUC values for susceptibility and risk assessment results obtained with slope units were higher than those obtained with grid units. Based on the comparison of the susceptibility and risk assessment results under the slope unit evaluation system and the actual survey data, it was concluded that the geological hazard assessment results under the slope unit system were in good agreement with the actual situation, and, finally, it was concluded that the geological hazard assessment results under the slope unit system were more reasonable and accurate. Thus, a scientific basis has been

provided for the selection of evaluation units in large-scale regional geological hazard assessments undertaken for the prevention and control of geological hazards.

**Author Contributions:** Conceptualization, S.L. and J.Z.; methodology, S.L.; software, D.Y.; validation, J.Z.; formal analysis, J.Z. and S.L.; investigation, B.M.; resources, J.Z.; data curation, S.L., D.Y. and B.M.; writing—original draft preparation, S.L.; writing—review and editing, S.L. and J.Z.; visualization, S.L.; supervision, J.Z. All authors have read and agreed to the published version of the manuscript.

**Funding:** This research received no external funding.

**Institutional Review Board Statement:** Not applicable.

**Informed Consent Statement:** Not applicable.

**Data Availability Statement:** Due to the nature of this research, the participants in this study did not agree for their data to be shared publicly, so supporting data are not available.

**Acknowledgments:** The authors would like to thank the editor and anonymous reviewers for their comments and suggestions which helped significantly to improve this paper.

**Conflicts of Interest:** The authors declare no conflict of interest.

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
