# Peer review of "Comparative Study of Geological Hazard Evaluation Systems Using Grid Units and Slope Units under Different Rainfall Conditions"

_sustainability, doi:10.3390/su142316153_

Round 1

Reviewer 1 Report

This paper studied the influences of two different evaluation units on the accuracy of evaluation results of the developed geological hazard evaluation system. This research is rich in content, but it is quite verbose in expressions and has some problems. The comments that should be addressed are listed as follows.

1. Through the whole article, the expressions are quite verbose. For example, the abstract should be revised to be more concise.

2. What does the abbreviation “ROC” and "AUC"mean in Line 29-30? The author should explain the meanings of abbreviations when first using them.

3. Line 42, this expression is not satisfactory, the evaluation units can not "determine" the scientificity and accuracy of evaluation results. It "influences" it.

4. Line 217-218, typo of double "information", the author should carefully check the whole manuscript to avoid this type of mistake.

5. Line 279-281, how does the evaluation model take into account these factors including geological structure, stratigraphic lithology and water system is not clear.

6. Section 5.2 lacks sufficient analysis of test results, a large part of this section just simply retells the data presented in Table 2 and Table 3. Besides, it is not clear what standard did the authors take to evaluate the accuracy of evaluation results from the two different evaluation units to come out the conclusion presented in line 448-451 and line 527-530.

Reviewer 2 Report

I review the manuscript titled "Comparative Study on Geological Hazard Evaluation System of Grid Unit and Slope Unit under Different Rainfall Conditions'. The comments are as follows which the dear authors should response and do. In my opinion the manuscript needs major revision.

1- Improve abstract part. The abstract should contain objectives, methods/analysis, findings, and novelty /improvement.

2- The highlights and important study results should be added in the introduction.

3- At the end of the introduction, introduce the novelty of your study and mention the organization of your paper.

4- Can the authors provide better quality for Figures 17 and 18? Unclear

5- Please re-write the Conclusion section.

6. Please add equation number to equations in line 178, 229, 241, 264.

Reviewer 3 Report

This study presents the influence of grid units and slope units for evaluating geological hazard system. The manuscript might draw attention of researchers who are interested in geological hazard evaluation. I recommend major revisions. The authors should address the following comments and revise their manuscript accordingly.

1.      Line 39: The authors shall highlight the differences between this study and previous studies and show the significance of this study.

2.      Line 39: It is suggested to review more previous studies in the relatively more outstanding SCI journals in the literature.

3.      Line 158: What does DEM mean? Please also check all the other abbreviations.

4.      In the discussion part, please also compare the results with the previous studies to enhance the significance and effectiveness of this study.

5.      Line 603: It is suggested to briefly describe the conclusions.

Reviewer 4 Report

Dear authors,

please revise all minor mistakes"

Abstract: re-format, un-bold

Line 39: the title should be from Number 1, not Zero

Line 59: wrong format, should be citation by number

Line 178, all equations need to mention in the relevant content; please check through the paper

Figure 3: unclear; please replace with the sharp photo, can not see the latitude and longitude, and the legend is unclear.

the conclusion: Line 604: "information" was repeated in this sentence; delete it. The authors need to rewrite; there are many sentences that are unclear, and some sentences need to be split into two sentences.

Round 2

Reviewer 2 Report

Dear Editor,
Thank you very much for your invitation to review the revised manuscript.
The revised manuscript is improved and the comments of reviewers were sufficiently responded and/or addressed.
Regards,

Reviewer 3 Report

Authors have generally addressed the comments.
